# Body Weight Loss Experience Among Adults from Saudi Arabia and Assessment of Factors Associated with Weight Regain: A Cross-Sectional Study

**DOI:** 10.3390/nu17142341

**Published:** 2025-07-17

**Authors:** Ibrahim M. Gosadi

**Affiliations:** Department of Family and Community Medicine, Faculty of Medicine, Jazan University, Jazan 45142, Saudi Arabia; gossady@hotmail.com

**Keywords:** body weight, healthy eating, physical activity, weight loss, Jazan, Saudi Arabia

## Abstract

**Background/Objectives**: Weight loss and its subsequent regain pose significant challenges for those dealing with overweight and obesity. This study explores weight loss strategies among adults in Saudi Arabia and evaluates factors linked to weight regain. **Methods**: This cross-sectional study focused on adults residing in Jazan, located in southwest Saudi Arabia. Data collection was conducted using a self-administered questionnaire that assessed participants’ demographics, medical history, perceptions of body weight, weight loss methods, and the incidence of weight regain. Logistic regression was used to determine whether there were statistically significant differences related to the occurrence of weight regain. **Results**: A total of 368 participants reported efforts to lose weight over the past 3 years. The average age of these participants was 32.7 years (standard deviation: 11.3), and the gender distribution was almost equal. The majority of the sample (65%) voiced dissatisfaction with their body weight. Some participants employed a combination of weight loss methods, with exercise, reduced food intake, and intermittent fasting being the most frequently mentioned. The findings also indicate that a minority sought professional help, whether from a physician or a nutritionist. Over 90% claimed to have successfully lost weight at least once during their attempts, but more than half (139 individuals) experienced weight regain following their weight loss efforts. Within the univariate logistic regression, higher odds ratios of weight regain were detected among men, older participants, those living in rural areas, individuals with higher levels of education, employed persons or business owners, those with higher monthly incomes, smokers, khat chewers, and those diagnosed with a chronic condition (*p* values < 0.05). However, the multivariate logistic regression revealed that only residence, monthly income, smoking status, and being diagnosed with a chronic disease remained statistically significant as predictors of weight regain after adjusting for other variables (*p* values < 0.05). **Conclusions**: These findings highlight the significance of incorporating weight regain prevention into body weight management for individuals dealing with overweight and obesity. Further research is needed to evaluate specific dietary, physical activity, and psychological factors that may increase the risk of weight regain in certain participants.

## 1. Introduction

Increased body weight presents a significant challenge affecting both individual and public health. The rising global prevalence and increasing trends over recent decades, along with the complexities of weight management related to genetic, social, and environmental influences, underscore the necessity of addressing overweight and obesity as a critical healthcare priority. Additionally, the links between excessive weight and various chronic diseases, as well as their effects on life expectancy, make managing overweight and obesity a pressing concern for both individuals and public health institutions.

Recent estimates from the World Health Organization indicate that overweight and obesity are significant factors affecting global health. By 2022, approximately two and a half billion adults were estimated to be overweight, with nearly 890 million of them suffering from obesity [1]. Furthermore, the growing prevalence of childhood obesity could result in higher rates of obesity among adults in the coming years unless effective prevention and control measures are implemented [2,3,4]. While countries may exhibit variations in the incidence of overweight and obesity, both conditions remain significant risk factors that increase the likelihood of morbidity and early mortality on a global basis [5].

Overweight and obesity are associated with multiple health complications and may even have a subsequent impact on socioeconomic indicators. According to the World Obesity Federation, it is predicted that the inability to treat or prevent obesity may result in an impact of USD 4.3 trillion by the year 2035 [6]. In a study that assessed the impact of obesity on different countries, it was noted that the United States and China are affected by the highest economic impact due to obesity in comparison to other countries [7]. Furthermore, in a study that assessed the impact of obesity on the gross domestic product (GDP), it was reported that obesity costs 0.8% of the GPD in India and 2.4% in Saudi Arabia [8].

Individuals with excess weight can be at risk of type two diabetes, cardiovascular diseases, and cancers, as well as a lower quality of life [9]. Those with obesity are more likely to incur higher costs of healthcare by 30% in comparison to individuals without obesity [10]. Obesity and its complications can incur indirect costs due to overall lower levels of activity leading to a reduction in productivity [11]. This rise in healthcare costs due to obesity is due to its association with subsequent comorbidities. For example, in a study that assessed the burden of obesity on related comorbidities in four European countries, it was indicated that chronic kidney diseases and cardiovascular diseases represent the main portion of financial costs due to the required tertiary care cost needed to manage these conditions [12].

Saudi Arabia, a high-income nation, is significantly impacted by rising rates of overweight, obesity, and chronic non-communicable diseases. The Saudi Arabian World Health Survey report indicates that less than 40% of adults in the country maintain a normal body weight. The prevalence of overweight and obesity among Saudi adults has approached around 60% [13]. The rise in body weight abnormalities is linked to an obesogenic environment, a Westernized lifestyle, dietary habits, and reduced physical activity [14,15,16,17]. In a systematic review that assessed the obesity burden in Saudi Arabia, it was indicated that obesity is related to an increased risk of hypertension, type two diabetes, and hypercholesterolemia [18]. Furthermore, according to a cost-of-illness analysis in Saudi Arabia, the economic burden of obesity in the country was estimated to reach USD 109 billion in 2022 from a healthcare services perspective, where treating the comorbidity of obesity was more costly than treating obesity itself [19].

To address the growing rates of overweight and obesity in Saudi Arabia, the Saudi Ministry of Health has initiated several programs and initiatives aimed at both professional and public health levels. A key initiative is the Obesity Control Program, which produced the Saudi guidelines on the prevention and management of obesity, published in 2016 [20,21]. The guidelines focus on the early detection of cases by assessing risk and encouraging preventive behaviors, such as healthy eating and physical activity. They also establish clear guidelines for therapy and bariatric surgery for high-risk individuals.

One recent example of the efforts made by government institutions in Saudi Arabia to combat rising overweight and obesity rates is the launch of a nationwide campaign called “Walk 30.” This campaign aims to encourage residents of Saudi Arabia to adopt a healthy and physically active lifestyle [22]. Additionally, recent reports show that pharmacological treatments and bariatric surgery are becoming more popular among adults in Saudi Arabia because of their swift effects on individual obesity management [23,24,25]. However, awareness and attitudes toward obesity interventions may vary among individuals, such as the fear associated with undergoing bariatric surgery [26].

Growing awareness about obesity’s impact on health might encourage some individuals struggling with obesity to pursue weight loss. A systematic review and meta-analysis evaluating the prevalence of weight control efforts among adults found that approximately 42% of adults attempt to lose weight. The review also identified various strategies for weight loss, including dietary changes, restrictions, seeking professional assistance, and, in certain cases, resorting to extreme measures such as fasting or vomiting [27].

The rising prevalence of overweight and obesity, along with the corresponding weight loss efforts among adults, underscores the need to evaluate weight loss strategies. Additionally, it is crucial to understand the factors linked to weight management and the prevention of weight regain. Jazan region is one of the regions affected by the rise in excess weight in the country where the prevalence of overweight and obesity reached 47.6% among adults [28]. The increased proportion of individuals with excess weight might motivate some to adopt weight loss measures and might also be at risk of weight regain. This study explores weight loss strategies among adults in Jazan, Saudi Arabia, and assesses the factors contributing to weight regain in this group.

## 2. Materials and Methods

### 2.1. Study Design and Settings

This assessment was a cross-sectional study conducted among adults residing in Jazan, a city in southern Saudi Arabia, to examine the body mass index (BMI) profile of the local population and the factors associated with body weight management. Participation was offered to the individuals online after they provided consent. The analysis included adults in the region who were attempting to lose weight, with no demographic or clinical exclusion criteria. Those who were approached but declined to participate were excluded from the recruitment process. Additionally, those who did not experience any attempts of weight loss were excluded from the current analysis. Ethical approval for the study was granted by the Standing Committee for Scientific Research of Jazan University (reference number REC-44/06/446, dated December 2022). Data collection took place between November 2023 and May 2024.

### 2.2. Data Collection Tool

Data was gathered using a self-administered questionnaire designed to assess various factors related to participants’ demographics, their diagnosed medical history related to body weight changes, perceptions about body weight, weight loss methods, and beliefs regarding factors affecting body weight control. Participants provided their body weight and height, enabling the estimation of their BMI. They were also surveyed about their attempts to lose weight over the past 3 years and how often they succeeded in losing weight. The participants were able to select the weight loss method according to their weight loss experience. Among the noted weight loss strategies, they indicated which method they found most effective for achieving their weight loss goals. Additionally, participants discussed their experiences with weight regain. The face validity of the questionnaire was established through pilot testing with a sample of 10 men and 10 women, which evaluated item clarity and the time required to complete the questionnaire. Content validity was assured through a review of the relevant literature concerning weight loss measures in a Saudi Arabian context. The Saudi Guidelines for Prevention and Management of obesity were consulted to identify weight loss measures as per the recommended healthcare practices. Additionally, similar literature assessing weight loss methods in the region of Jazan was also consulted to identify weight loss methods practiced in the same community, which might be considered harmful or less healthier options for weight loss. The compiled list of weight loss measures included exercising, consuming a lower amount of food, intermittent fasting, selecting low-calorie food items, the use of artificial sweeteners, consulting a nutritionist or a physician, the use of medications associated with weight loss, purging, or bariatric surgery [20,29].

### 2.3. Data Collection Process

The questionnaire was created online using Google Forms. The identification and engagement stages of data collection were initiated by generating a web link, which was then shared on relevant social media platforms, primarily WhatsApp, to connect with the target participants. An information sheet describing the study’s purpose and procedures accompanied the form. Those who agreed to participate gained access to the questionnaire, while those who declined were redirected to an alternative location.

A convenient, non-random sampling method was employed to reach the targeted participants, and those who participated were encouraged to share the web link with their contacts. At the time of this study, limited data existed regarding weight loss attempts among adults in Saudi Arabia. However, the estimation of sample size focused on individuals with abnormal body weight who were likely to pursue weight loss. The StatCalc feature of Epi Info was employed to calculate the sample size for this research. Based on estimates from the Saudi Arabian World Health Survey, which indicates that almost 60% of adults in Saudi Arabia are overweight or obese [13], a sample size of 637 participants was determined necessary to evaluate BMI abnormalities and related weight loss attempts, assuming a 60% frequency, a 99% confidence level, and a 5% margin of error.

### 2.4. Data Analysis

Data analysis was conducted using the Statistical Package for Social Sciences, version 25. Descriptive statistics involved calculating frequencies and proportions for binary and categorical variables, as well as means and standard deviations for continuous variables. Cross-tabulation was used to assess variations in factors related to the occurrence of weight regain after weight loss. The Chi-squared test estimated the presence of statistically significant differences among the variables studied. A *p*-value of 0.05 or lower indicated statistical significance for the tests applied. For cross-tabulation purposes, age was categorized based on the sample mean, distinguishing between individuals younger than 32 and those aged 32 or older. Furthermore, to investigate employment effects and minimize empty cells in the cross-tabulation, individuals with government or private jobs and business owners were grouped, while those not employed at the time of data collection, such as retirees, unemployed individuals, or homemakers, were classified into a separate group. The cross-tabulation bivariate analysis was followed up by univariate logistic regressions to estimate the direction and effect size of each independent variable on weight regain risk. Finally, the univariate logistic regression was followed up by multivariate logistic regression to identify the predictors associated with weight regain risk with statistical significance while adjusting for the influence of other variables in the current sample.

## 3. Results

A total of 798 people responded to the invitation for the current assessment, with 368 (46%) reporting attempts to lose weight in the past 3 years. The demographic characteristics of the participants included in this analysis are shown in Table 1. The average age of the participants was 32.7 years (standard deviation: 11.3), with an almost balanced gender distribution. Most participants (59%) reported living in urban areas, and 73% had completed an undergraduate degree at a university or college. Regarding employment, the most common responses were governmental employees (37%) and students (34%). Furthermore, most respondents reported a monthly income of less than SAR 5000. When asked about their smoking and khat chewing habits, the majority stated they never smoked or chewed khat. Finally, the estimated BMI levels in this sample indicate a high prevalence of overweight and obesity, with only 28% identified as normal weight, 34% classified as overweight, and 29% as obese.

Table 2 presents the frequencies and proportions of diagnosed medical conditions in the sample. Participants could report multiple morbidities. The most common condition is obesity, affecting 18% of the sample, which is anticipated given the targeted group. Dyslipidemia, hypertension, diabetes, and blood disorders were reported by over 5% of participants. Additionally, only 193 individuals (52%) in the sample reported no diagnosed conditions.

Table 3 presents an overview of participants’ perceptions of their body weight, motivations for weight loss, the number of attempts made to lose weight, and their success in achieving this goal. The majority of the sample (65%) expressed dissatisfaction with their body weight, and most perceived themselves as either overweight or extremely overweight (77%). When asked about their motivation for losing weight, the predominant response was to reduce the risk of disease, followed by the desire for an improved body image. The results further reveal that nearly 60% of participants attempted to lose weight three or more times in the past 3 years. More than 90% indicated they managed to lose weight, at least during some of their attempts. However, over half (139) reported experiencing weight regain following their weight loss efforts.

Figure 1 shows the weight loss methods used by participants (in blue) alongside their opinions on which methods successfully facilitated their weight loss (in red). Many participants reported using a combination of several approaches, with exercise, reduced food intake, and intermittent fasting being the most common. Participants also identified these methods as effective for weight loss. Additionally, focusing on low-calorie food items and opting for artificial sweeteners instead of high-calorie options was a significant factor in achieving weight loss. The findings suggest that a minority sought professional help, whether from a physician or a nutritionist, while some individuals chose to use herbal medications or medications without consulting a healthcare professional. Induced vomiting and bariatric surgery were the least mentioned methods.

Figure 2 illustrates the participants’ views on factors that may assist with their weight loss efforts. The most commonly reported supportive elements included having a caring family, supportive friends, and public facilities that encourage exercise. A small portion of the sample acknowledged that the availability and cost of healthy food are supportive factors, which indicates that these aspects are concerns for the majority of the participants.

Table 4 presents demographic factors associated with weight regain within the sample. Weight regain was observed more frequently among men, older individuals, those living in rural areas, individuals with higher education, employed individuals or business owners, those with higher monthly incomes, smokers, khat chewers, and those diagnosed with a clinical condition (*p* values < 0.05). The detection of significant associations between these variables suggests that aging, employment status, an unhealthy lifestyle, and being diagnosed with a condition are critical risk factors for weight regain.

Table 5 and Table 6 represent the findings of the applied logistic regression. The univariate logistic regression indicates the presence of higher odds of weight regain among males, those who are older than 32, those living in urban areas, those weight higher education degrees, employed, those with higher income, smokers, those who are previous khat chewers, and individuals diagnosed with a chronic disease (*p* values < 0.05). The strongest effects were observed among those with a postgraduate degree (OR: 4.1 95% CI: 1.54–10.8), higher income (OR: 3.1, 95% CI: 1.72–5.59), and smokers (OR 2.31 95% CI: 1.26–4.24). However, after applying the multivariate regression analysis to assess which variables statistically contribute to the prediction of weight regain while adjusting for other variables, it can be noted that only residence, monthly income, smoking status, and being diagnosed with a chronic disease remained statistically significant (Table 6).

## 4. Discussion

This study was a cross-sectional investigation aimed at assessing weight loss methods and factors associated with weight regain among adults. The majority of the sample was either overweight or obese and not satisfied with their body weight. Most participants had made several attempts to lose weight over the last 3 years, with disease prevention being the main motivator. Although many were able to lose weight, half of them regained it. Factors associated with weight regain included aging, male gender, employment, smoking, khat chewing, and having a diagnosed medical condition.

The results of this investigation align with findings from other local and international studies. A study on weight loss among 501 adolescents in Saudi Arabia revealed that the primary methods used for weight loss included reducing food intake, exercising, and fasting [30]. This aligns with current findings indicating that the majority of the sample predominantly utilized these three methods. Although there are differences in age groups between adolescents and adults, these methods appear to be preferred strategies for weight loss.

A similar study in Saudi Arabia by Alfadda et al. focused on individuals with obesity (BMI of ≥30) to assess their motivation and experiences regarding weight loss. It was found that 92% of participants attempted to lose weight. This figure is significantly higher compared to the current survey’s 46% weight loss attempts, which is understandable considering that the current participants had normal body weight. Furthermore, Alfadda et al. revealed that among those who did lose weight, only 5% were able to sustain their weight loss for over a year [31]. Our assessment did not measure the duration of weight regain. However, this suggests that sustaining body weight after loss is challenging for individuals with obesity. Consequently, any weight loss program should incorporate measures to prevent weight regain, as depending solely on weight loss strategies is insufficient.

International assessments provide additional evidence of the challenges in maintaining body weight following weight loss. For example, a systematic review encompassing 32 studies on weight regain following bariatric surgery found that 1 in 6 patients tend to regain weight. This risk of regaining weight is associated with surgical, genetic, dietary, and psychiatric factors [32]. It is worth noting that the majority of weight regain assessments are conducted among patients who undergo bariatric surgeries, while the evidence of weight regain related to other weight loss options remains limited.

The current study identified specific demographic factors associated with weight regain, including gender, age, and education. An Italian study involving 64 postmenopausal women concluded that 50% of participants regained weight, with lower education levels and unhealthy eating habits identified as contributing factors [33]. The current analysis did not evaluate weight regain in relation to dietary habits. However, our sample suggests that weight regain was more pronounced among educated individuals, likely linked to employment status. A separate American study involving 1310 participants aged 20–84 found that being of Mexican ethnicity and leading a sedentary lifestyle were significant risk factors for weight regain, whereas gender, education level, and age showed no association with weight regain [34]. This may suggest that cultural differences among countries contribute to the overall variation in obesity risk or the risk of weight regain.

In this study, fewer than 11% of participants reported consulting a nutritionist or physician during their weight loss efforts. Moreover, 24 individuals reported using herbal medicine for weight loss, while 18 indicated that they took weight loss medications without a physician’s guidance. Only 15 participants (4%) reported using weight loss medications. A review focused on the use of anti-obesity medications in the Arab world concluded that liraglutide was the most researched drug. It also found that nearly one-third of users experienced gastrointestinal side effects from these medications [23]. Fear of side effects can influence the use of weight loss medications in the studied community.

Similarly to the underutilization of weight loss medications, only 10 participants reported undergoing bariatric surgery, and 7 of them view it as an effective weight loss solution. The small number of reported surgeries can be attributed to the eligibility criteria for weight loss. Nevertheless, studies examining awareness and attitudes towards bariatric surgery reveal limited knowledge and concerns about potential complications related to the procedure [26,35].

In the current study, of the 368 participants who attempted weight loss, only 88 (24%) expressed satisfaction with their body weight, while most still considered themselves overweight or obese. This belief may act as a motivation for weight loss in this group. A larger study involving 3411 residents of Saudi Arabia found that 71% were satisfied with their body weight, despite some having abnormal body weight, highlighting a potential satisfaction–behavior paradox [28].

It is worth noting that the current findings indicate the presence of certain characteristics associated with the risk of weight regain despite limited statistical significance in the multivariate logistic regression. For example, the lack of statistical significance concerning the impact of education in the multivariate analysis can be partially explained by the small sample size of individuals with higher education in comparison to other categories. Furthermore, the current findings suggest that individuals who are exposed to secondhand smoking might be at risk of weight regain with marginal statistical significance (OR 1.69 95% CI: 0.97–2.95). Individuals exposed to secondhand smoking might be associated with unhealthy lifestyle of families, friends, or peers. This indicates the importance of considering some factors associated with weight regain from a public health perspective even with the limited statistical significance and the importance of considering these notions as an area for further research.

The current study has several strengths and weaknesses. The main strength lies in its ability to assess weight loss attempts among individuals who have used various methods and to evaluate weight regain based on measured demographic characteristics. This can be beneficial for future targeted interventions, which should not only promote short-term weight loss practices but also encourage long-term, sustainable interventions. A significant limitation of the study is its subjective nature, primarily influenced by the participants’ perspectives, rather than associating findings with specific clinical measures. However, it can be argued that this approach is valuable, as only a small proportion of this community typically seeks professional healthcare when trying to lose weight. Another limitation is related to the generalizability of the current findings since it only targeted one region in Saudi Arabia. However, it can be postulated that similar cultural and socioeconomic backgrounds in all regions of Saudi Arabia is likely to impact lifestyle choices, including weight modification measures. The current study has a relatively limited sample size, due to the nature of the research question which only targeted those who attempted to lose weight during the last three years, leading to the exclusion of many participants and subsequently limiting the application of certain statistical tests (such as stratification by age, gender, and social stratum). Therefore, future studies are recommended to include larger samples and to involve other regions in the country.

## 5. Conclusions

The majority of the recruited participants were either overweight or obese and dissatisfied with their body weight, which may have motivated them to attempt weight loss. However, regaining weight poses a significant challenge that requires further investigation. Certain socioeconomic factors, lifestyle, and being affected with a chronic disease increase the odds of weight regain. These findings have clinical implications, suggesting the need to target subjects with higher income, diagnosed with a chronic condition and smokers with tailored strategies that suit their specific needs. Further research is needed to investigate specific dietary, physical activity, and psychological factors that may increase the risk of weight regain for certain participants.

## Figures and Tables

**Figure 1 nutrients-17-02341-f001:**
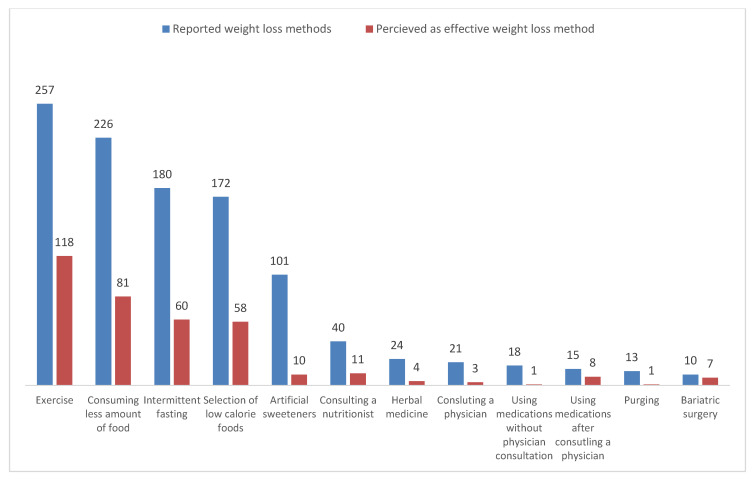
Weight loss methods among 368 participants from Jazan, Saudi Arabia.

**Figure 2 nutrients-17-02341-f002:**
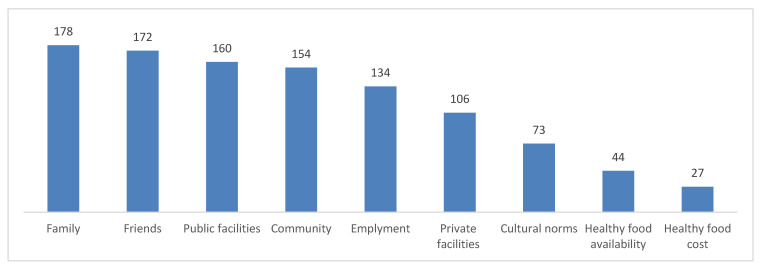
Factors supporting weight loss attempts among 368 participants from Jazan, Saudi Arabia.

**Table 1 nutrients-17-02341-t001:** Demographic characteristics of 368 participants from Jazan, Saudi Arabia, who attempted to lose weight.

Variables	Frequency [Proportion]
Age *	
Younger than 32	180 [49%]
32 or older	187 [51%]
Gender	
Male	186 [50.5%]
Female	182 [49.5%]
Area of residence	
Rural	159 [43.2%]
Urban	209 [56.8%]
Education level	
School education	69 [18.7%]
Undergraduate university	270 [73.4%]
Postgraduate University	29 [7.9%]
Employment *	
Governmental employee	136 [37.1%]
Military Employee	18 [4.9]
Private sector employee	30 [8.2%]
Retired	15 [4.1%]
Unemployed	19 [5.2%]
Business Owner	3 [0.8%]
Student	125 [34%]
Housewife	21 [5.7%]
Monthly income	
Less than SAR 5000	148 [40.2%]
Between SAR 5000 and 10,000	54 [14.8%]
More than SAR 10,000 and less than SAR 150,000	89 [24.2%]
More than SAR 150,000	76 [20.8%]
Smoking	
Currently smoking	59 [16%]
Previously smoking	24 [6.5%]
Secondhand smoking	77 [20.9%]
Never smoker	208 [56.6%]
Khat chewing	
Current	15 [4.1%]
Previous	44 [12.2%]
Never	308 [83.7%]
Current BMI category *	
Underweight	23 [6.4%]
Normal weight	103 [28.7%]
Overweight	125 [34.8%]
Obese	108 [30.1%]

* One missing for age, one missing for employment, nine missing for BMI category.

**Table 2 nutrients-17-02341-t002:** Frequency of reported diagnosed conditions among 368 participants from Jazan, Saudi Arabia, who attempted to lose weight.

Variables	Frequency [Proportion]
Diabetes	23 [6.3%]
Dyslipidemia	38 [10.3%]
Hypertension	40 [10.9%]
Obesity	67 [18.2%]
Blood disorders	23 [6.3%]
Sleep disorder	17 [4.6%]
Asthma	18 [4.9%]
Mental illness	8 [2.2%]
Hormonal condition	15 [4.1%]
Dental condition	21 [5.7%]
GIT condition	18 [4.9%]
Cancer	1 [0.3%]
Menopause	2 [0.5%]
History of pregnancy	62 [34.3% *]

* Proportion among female participants.

**Table 3 nutrients-17-02341-t003:** Satisfaction with and perceptions of body weight among 368 participants from Jazan, Saudi Arabia, who attempted to lose weight.

Statement	Frequency [Proportion]
Are you satisfied with your current body weight?	
Strongly satisfied	18 [4.95%]
Satisfied	70 [19%]
I do not know	38 [10.3%]
Unsatisfied	154 [41.85%]
Strongly unsatisfied	88 [23.9%]
Which of the following is your opinion about your body weight?	
I have a low body weight	4 [1.1%]
I have a normal body weight	81 [22%]
I am overweight	254 [69%]
I am extremely overweight	29 [7.9%]
Reasons for losing weight	
Diseases prevention	305 [82.9%]
To improve body image	277 [75.3%]
To feel proud with a good weight	139 [37.8%]
To raise my self confidence	212 [57.6%]
Number of weight loss attempts during the last 3 years *	
Once	87 [23.7%]
Twice	70 [19.1%]
Three times	85 [23.2%]
Four times	24 [6.5%]
Five times	16 [4.3%]
More than five times	85 [23.2%]
Ability to lose weight *	
Yes, with every attempt	207 [56.4]
Yes, during some attempts	130 [35.4%]
No	30 [8.2%]
Weight regain	193 [52.4%]

* One missing case.

**Table 4 nutrients-17-02341-t004:** Demographic factors associated with weight regain following weight loss among 368 participants from Jazan, Saudi Arabia.

Variables	Weight Regain Frequency [Proportion]			
	Yes	No	Total	*p* Value
Gender				0.005
Male	111 [59.7%]	75 [40.3%]	186 [100%]	
Female	82 [45.1%]	100 [54.9%]	182 [100%]	
Age				0.015
Younger than 32	83 [46.1%]	97 [53.9%]	180 [100%]	
32 or older	110 [58.8%]	77 [41.2%]	187 [100%]	
Residence				0.008
Rural	96 [60.4%]	63 [39.6%]	159 [100%]	
Urban	97 [46.4%]	112 [53.6%]	209 [100%]	
Education				0.014
School	30 [43.5%]	39 [56.5%]	69 [100%]	
Undergraduate	141 [52.2%]	129 [47.8%]	270 [100%]	
Postgraduate	22 [75.9%]	7 [24.1%]	29 [100%]	
Employment				0.003
Employed or business owner	114 [61%]	73 [39%]	187 [100%]	
Unemployed or retired	23 [41.8%]	32 [58.2%]	55 [100%]	
Student	55 [44%]	70 [56%]	125 [100%]	
Monthly income				0.001
Less than SAR 5000	63 [42.6%]	85 [57.4%]	[100%]	
Between SAR 5000 and SAR 10,000	26 [48.1%]	28 [51.9%]	54 [100%]	
More than SAR 10,000 and less than SAR 15,000	50 [56.2%]	39 [43.8%]	89 [100%]	
SAR 15,000 or more	53 [69.7%]	23 [30.3%]	79 [100%]	
Smoking				0.021
Current	39 [66.1%]	20 [33.9%]	59 [100%]	
Previous	14 [58.3%]	10 [41.7%]	24 [100%]	
Second hand	45 [58.4%]	32 [41.6%]	77 [100%]	
Never	95 [45%]	113 [54.3%]	208 [100%]	
Khat chewing				0.015
Current	11 [73.3%]	4 [26.7%]	15 [100%]	
Previous	30 [68.2%]	14 [31.8%]	44 [100%]	
Never	151 [49.1%]	157 [50.9%]	308 [100%]	
Diagnosed condition				0.001
No [healthy]	85 [44%]	108 [56%]	193 [100%]	
Yes [diagnosed]	108 [61.7%]	67 [38.3%]	175 [100%]	

**Table 5 nutrients-17-02341-t005:** Univariate logistic regression of variables associated with the odds of weight regain following weight loss among 368 participants from Jazan, Saudi Arabia.

Independent Variables	Odds Ratio for Weight Regain		
	OR	95% CI	*p* Value
Gender			
Female	Reference		
Male	1.80	1.19–2.72	0.005
Age			
Younger than 32	Reference		
32 or older	1.66	1.10–2.52	0.015
Residence			
Urban	Reference		
Rural	1.75	1.15–2.67	0.008
Education			
School	Reference		
Undergraduate	1.42	0.83–2.42	0.195
Postgraduate	4.08	1.54–10.8	0.004
Employment			
Student	Reference		
Employed or business owner	1.98	1.25–3.14	0.003
Unemployed or retired	0.91	0.48–1.73	0.785
Monthly income			
Less than SAR 5000	Reference		
Between SAR 5000 and SAR 10,000	1.25	0.67–2.34	0.479
More than SAR 10,000 and less than SAR 15,000	1.72	1.01–2.94	0.042
SAR 15,000 or more	3.10	1.72–5.59	0.0001
Smoking			
Never	Reference		
Current	2.31	1.26–4.24	0.006
Previous	1.66	0.70–3.92	0.243
Second hand	1.67	0.98–2.83	0.056
Khat chewing			
Never	Reference		
Current	2.85	0.89–9.17	0.077
Previous	2.22	1.13–4.36	0.019
Diagnosed condition			
No [healthy]	Reference		
Yes [diagnosed]	2.04	1.35–3.10	0.001

OR: Odds ratio; 95% CI: 95% confidence interval.

**Table 6 nutrients-17-02341-t006:** Multivariate logistic regression of variables associated with the odds of weight regain following weight loss among 368 participants from Jazan, Saudi Arabia.

Independent Variables	Odds Ratio for Weight Regain		
	OR	95% CI	*p* Value
Residence			
Urban	Reference		
Rural	1.91	1.22–2.99	0.004
Monthly income			
Less than SAR 5000	Reference		
Between SAR 5000 and SAR 10,000	1.06	0.54–2.05	0.856
More than SAR 10,000 and less than SAR 15,000	1.64	0.93–2.89	0.082
SAR 15,000 or more	2.93	1.58–5.45	0.001
Smoking			
Never	Reference		
Current	2.16	1.13–4.14	0.019
Previous	1.56	0.63–3.88	0.334
Second hand	1.69	0.97–2.95	0.063
Diagnosed condition			
No [healthy]	Reference		
Yes [diagnosed]	1.97	1.27–3.07	0.002

OR: Odds ratio; 95% CI: 95% confidence interval.

## Data Availability

The data presented in this study are available on request from the corresponding author.

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
