# Peer review of "Body Weight Loss Experience Among Adults from Saudi Arabia and Assessment of Factors Associated with Weight Regain: A Cross-Sectional Study"

_nutrients, 2025, doi:10.3390/nu17142341_

Round 1
Reviewer 1 Report
Comments and Suggestions for Authors
The reviewer would like to point out a few items for the author's consideration below:
- Include more statistics and information on the subject from an epidemiological standpoint in the literature. Prevalence and cost of obesity globally and in Saudi Arabia, burden of disease etc
- What were the inclusion and exclusion criteria? How were they determined? How was the sample size determined?
- It is unclear if the questionnaires and surveys used were validated or constructed based on available literature for Saudi Arabia.
- Why was the 32 years mark used as a cutoff point? It is more typical in studies to use the 40yr mark.
- What was the rationale for sectioning the income strata the way presented?
- In the analysis is type 1 and type 2 diabetes considered together?
- Was there a particular reason as to why the geographic area of Jazan was selected for the study?
- Was there a combination of methods per attempts to lose weight? Often times when a person attempts to lose weight they do so combining more than one strategies.
- A stronger and more robust list of references should be compiled.
English is OK although the manuscript would benefit from an English native speaker read through.
Author Response
The author of the manuscript appreciates the important comments of the reviewer. Below are point by point responses to all raised comments. We believe that the applied modifications as per the instructions of the reviewer have enhanced the writing quality of the manuscript.
Comment 1: Include more statistics and information on the subject from an epidemiological standpoint in the literature. Prevalence and cost of obesity globally and in Saudi Arabia, burden of disease etc
Response: The author of the manuscript appreciate the comment of the reviewer. More statistics are now added to the introduction to provide more overview on obesity from an epidemiological standpoint. Additionally, prevalence and cost of obesity globally and in Saudi Arabia, and burden of the disease are now added to the introduction. These modification are highlighted in yellow in the revised manuscript as the following:
‘Recent estimates from the World Health Organization indicate that overweight and obesity are significant factors affecting global health. By 2022, approximately two and a half billion adults were estimated to be living with overweight, with nearly 890 million of them suffering from obesity [1]. Furthermore, the growing prevalence of childhood obesity could result in higher rates of obesity among adults in the coming years unless effective prevention and control measures are implemented [2-4]. While countries may exhibit variations in the incidence of overweight and obesity, both conditions remain significant risk factors that increase the likelihood of morbidity and early mortality on global basis [5].
Overweight and obesity are associated with multiple health complications and may even have a subsequent impact on socioeconomic indicators. According to the World Obesity Federation, it is predicted that inability to treat or prevent obesity, may result in an impact of 4.3 trillion US dollar by the year 2035 [6]. In a study that assessed impact of obesity on different countries, it was noted that the United States and China are affected by the highest economic impact due to obesity in comparison to other countries [7]. Furthermore, in a study that assessed the impact of obesity on the gross domestic product (GDP), it was reported that obesity costs 0.8% of the GPD in India and 2.4% in Saudi Arabia [8].
Individuals with excess weight can be at risk of type two diabetes, cardiovascular diseases, and cancers and lower quality of life [9]. Those with obesity are more likely to incur higher cost of healthcare by 30% in comparison to individuals without obesity [10]. Obesity and its complications, can incur indirect cost due to overall lower level of activity leading to reduction of productivity [11]. This rise in healthcare cost due to obesity is due to its association with subsequent comorbidities. For example, in a study that assessed the burden of obesity on related comorbidities in four European countries, it was indicated that chronic kidney diseases, and cardiovascular diseases represent main financial cost due to the required tertiary care cost needed to manage these conditions [12].
Saudi Arabia, a high-income nation, is significantly impacted by rising rates of overweight, obesity, and chronic non-communicable diseases. The Saudi Arabian World Health Survey report indicates that less than 40% of adults in the country maintain a normal body weight. The prevalence of overweight and obesity among Saudi adults has approached around 60% [13]. The rise in body weight abnormalities is linked to an obe-sogenic environment, a Westernized lifestyle, dietary habits, and reduced physical activity [14-17]. In a systematic review that assessed obesity burden in Saudi Arabia, it was indicated that obesity is related to increased risk of hypertension, type two diabetes, and hypercholesterolemia [18]. Furthermore, according to a cost-of-illness analysis in Saudi Arabia, economic burden of obesity in the country was estimated to reach 109 billion US dollars in 2022 from a healthcare services perspective, where treating the comorbidity of obesity was more costly than treating obesity itself [19].’
Comment 2: What were the inclusion and exclusion criteria? How were they determined? How was the sample size determined?
Response: The inclusion and exclusion criteria are detailed within the study design and settings section of the materials and methods. The criteria are highlighted with yellow within the revised manuscript as the following:
‘Participation was offered to the individuals online after they provided consent. The analysis included adults in the region who were attempting to lose weight, with no demographic or clinical exclusion criteria. Those who were approached but declined to participate were excluded from the recruitment process. Additionally, those who did not experience any attempts of weight loss were excluded from the current analysis.’
Sample size estimation is detailed within the data collection process section of the materials and methods. The estimation process is highlighted with yellow within the revised manuscript as the following:
‘A convenient, non-random sampling method was employed to reach the targeted participants, and those who participated were encouraged to share the web link with their contacts. At the time of this study, limited data existed regarding weight loss attempts among adults in Saudi Arabia. However, the estimation of sample size focused on individuals with abnormal body weight who were likely to pursue weight loss. The StatCalc feature of Epi Info was employed to calculate the sample size for this research. Based on estimates from the Saudi Arabian World Health Survey, which indicates that almost 60% of adults in Saudi Arabia are overweight or obese [13], a sample size of 637 participants was determined necessary to evaluate BMI abnormalities and related weight loss attempts, assuming a 60% frequency, a 99% confidence level, and a 5% margin of error.’
Comment 3: It is unclear if the questionnaires and surveys used were validated or constructed based on available literature for Saudi Arabia.
Response: We appreciate the comment of the reviewer. More details about the content of the questionnaire and validation are now added to the data collection tool section within the materials and methods of the revised manuscript. The following is highlighted with yellow in the revised manuscript as a response to the raised point:
‘Content validity was assured through a review of relevant literature concerning weight loss measures in a Saudi Arabian context. The Saudi Guidelines for Prevention and Management of obesity were consulted to identify weight loss measures as per the recommended healthcare practices. Additionally, similar literature assessing weight loss methods in the region of Jazan was also consulted to identify weight loss methods prac-ticed in the same community, which might be considered harmful or less healthier op-tions for weight loss. The compiled list of weight loss measures included exercising, consuming less amount of food, intermittent fasting, selection of low calorie food items, use of artificial sweeteners. consulting a nutritionist or a physician, use of medications associated with weight loss, purging, or bariatric surgery [20, 29].’
Comment 4: Why was the 32 years mark used as a cutoff point? It is more typical in studies to use the 40yr mark.
Response: We agree with the comment of the reviewer that marking a specific age as a cut-off point depends on context, population, and nature of the assessment. However, in the current investigation, the cut-off point was selected based on the calculated mean age of the sample. This process is highlighted with yellow within the data analysis section of the materials and methods as the following:
‘ For cross-tabulation purposes, age was categorized based on the sample mean, distinguishing between individuals younger than 32 and those aged 32 or older.’
Comment 5: What was the rationale for sectioning the income strata the way presented?
Response: The rational of sectioning the income data into four categories; less than 5000 SAR, between 5000 and 10000 SAR, more than 10000 and less than 150000 SAR, More than 150000 SAR, is due to the recent estimated by the Saudi General Authority for Statistics which indicates that the monthly average wage in Saudi Arabia is 10238 SAR. The income data in the current study was sectioned around 10000 SAR as a median cut-off point. Asking about lower or higher income category was necessary because of the expected socioeconomic variability of the targeted sample. More information about the income data can be found at the following webpage of the General Authority for Statistics:
https://www.stats.gov.sa/en/w/gastat-saudi-workers-monthly-average-wage-in-four-sectors-10.238-sar
No modification was applied to the revised manuscript in response to this particular comment.
Comment 6: In the analysis is type 1 and type 2 diabetes considered together?
Response: We appreciate the important comment of the reviewer. We acknowledge that type 1 and type 2 diabetes have different pathophysiology with different impact on the body weight. However, for the current analysis, and due to the limited sample size, only 23 patients reported being diagnosed with diabetes (regardless to the type). Within the bivariate analysis, all diagnosed conditions (including diabetes) were grouped into one variable (being diagnosed with a chronic disease or not) to assess its association with the weight regain which was statistically significant (P value 0.001), suggesting that having a diagnosed condition poses higher risk of weight regain in comparison to healthy individuals.
No modification was applied to the revised manuscript in response to this particular comment.
Comment 7: Was there a particular reason as to why the geographic area of Jazan was selected for the study?
Response: We appreciate the important comment of the reviewer. We agree that expanding the recruitment to a larger geographical area to include other regions in Saudi Arabia is more likely to enhance the generalizability of the findings. However, targeting the region of Jazan was decided due to logistic reasons as this study was not funded. Nonetheless, the region, as many other regions in Saudi Arabia, suffers from increasing prevalence of overweight and obesity where 47% residents are either overweight or obese [1] , which is similar to excess weight prevalence in the country.
- Makeen, A.M.; Gosadi, I.M.; Jareebi, M.A.; Muaddi, M.A.; Alharbi, A.A.; Bahri, A.A.; Ryani, M.A.; Mahfouz, M.S.; Al Bahhawi, T.; Alaswad, A.K.; et al. Satisfaction-Behavior Paradox in Lifestyle Choices: A Cross-Sectional Study of Health Behaviors and Satisfaction Levels in Jazan, Saudi Arabia. Healthcare 2024, 12, 1770. https://doi.org/10.3390/healthcare12171770
To indicate the importance of conducting this study in the region, the following was added to the last paragraph of the introduction to indicate rational of conducting the study in the Jazan region. The section was highlighted with yellow in the introduction of the revised manuscript as the following:
‘ The rising prevalence of overweight and obesity, along with the corresponding weight loss efforts among adults, underscores the need to evaluate weight loss strategies. Additionally, it is crucial to understand the factors linked to weight management and the prevention of weight regain. Jazan region is one of the regions affected by the rise of excess weight in the country where the prevalence of overweight and obesity reached 47.6% among adults [28]. The increased proportion of individuals with excess weight might motivate some to adopt weight loss measure and might also be at risk of weight regain. This study explores weight loss strategies among adults in Jazan, Saudi Arabia, and assesses the factors contributing to weight regain in this group.’
We also added the following to the last paragraph of the discussion section of the revised manuscript to indicate the importance of targeting larger sample size from other regions in the country to enhance the generalizability. The section is highlighted with yellow as the following:
‘Another limitation is related to the generalizability of the current findings since it only targeted one region in Saudi Arabia. However, it can be postulated that similar cultural, and socioeconomic background in all region of Saudi Arabia is likely to impact lifestyle choices, including weight modification measures. The current study has a relatively limited sample size, due to the nature of the research question which only targeted those who attempted to lose weight during the last three years leading to exclusion of many participants and subsequently limiting the application of certain statistical tests (such as stratification by age, gender, and social stratum). Therefore, future studies are recommended to include larger sample and to involve other regions in the country.’
Comment 8: Was there a combination of methods per attempts to lose weight? Often times when a person attempts to lose weight they do so combining more than one strategies.
Response: We agree with the comment of the reviewer that some might use a combination of weight loss method to reach their desired body weight. This was identified in our findings and was illustrated within figure one and the text which states that some used a combination of weight loss methods. To enhance the writing clarity of the manuscript, the following was added to the data collection tool section of the methodology:
‘The participants were able to select more than weight loss method according to their weight loss experience.’
Additionally, the following statement is made in the results section to report the use of more than weight loss method among the recruited sample:
‘Many participants reported using a combination of several approaches, with exercise, reduced food intake, and intermittent fasting being the most common. Participants also identified these methods as effective for weight loss. Additionally, focusing on low-calorie food items and opting for artificial sweeteners instead of high-calorie options was a significant factor in achieving weight loss.’
The indicated sections are highlighted with yellow in the revised manuscript to allow easier assessment of the modifications.
Comment 9: A stronger and more robust list of references should be compiled.
Response: a stronger and more robust list of references is now used. The following references were added during the revision stage:
- Ng M, Gakidou E, Lo J, Abate YH, Abbafati C, Abbas N, et al. Global, regional, and national prevalence of adult overweight and obesity, 1990–2021, with forecasts to 2050: a forecasting study for the Global Burden of Disease Study 2021. The Lancet. 2025;405(10481):813-38.
- World Obesity Federation. Economic impact of overweight and obesity to surpass $4 trillion by 2035 2023 [Available from: https://www.worldobesity.org/news/economic-impact-of-overweight-and-obesity-to-surpass-4-trillion-by-2035. Accessed on 5th of July 2025.
- Sweis NJ. The economic burden of obesity in 2024: a cost analysis using the value of a statistical life. Critical Public Health. 2024;34(1):1-13.
- Okunogbe A, Nugent R, Spencer G, Ralston J, Wilding J. Economic impacts of overweight and obesity: current and future estimates for eight countries. BMJ global health. 2021;6(10).
- Wold Health Organization. Obesity and overweight 2024 [Available from: https://www.who.int/news-room/fact-sheets/detail/obesity-and-overweight. Accessed on 5th of July 2025.
- Withrow D, Alter DA. The economic burden of obesity worldwide: a systematic review of the direct costs of obesity. Obes Rev. 2011;12(2):131-41.
- Yusefzadeh H, Rashidi A, Rahimi B. Economic Burden of Obesity: A Systematic Review. Asian Journal of Social Health and Behavior. 2019;2(1).
- Athanasakis K, Bala C, Kokkinos A, Simonyi G, Karoliová KH, Basse A, et al. The economic burden of obesity in 4 south-eastern European countries associated with obesity-related co-morbidities. BMC Health Serv Res. 2024;24(1):354.
- Al-Omar HA, Alshehri A, Alqahtani SA, Alabdulkarim H, Alrumaih A, Eldin MS. A systematic review of obesity burden in Saudi Arabia: Prevalence and associated co-morbidities. Saudi Pharmaceutical Journal. 2024;32(11):102192.
- Makeen AM, Gosadi IM, Jareebi MA, Muaddi MA, Alharbi AA, Bahri AA, et al. Satisfaction-Behavior Paradox in Lifestyle Choices: A Cross-Sectional Study of Health Behaviors and Satisfaction Levels in Jazan, Saudi Arabia. Healthcare [Internet]. 2024; 12(17).
- Nagi MA, Almalki ZS, Thavorncharoensap M, Sangroongruangsri S, Turongkaravee S, Chaikledkaew U, et al. The Burden of Obesity in Saudi Arabia: A Real-World Cost-of-Illness Study. ClinicoEconomics and outcomes research : CEOR. 2025;17:233-46.
Comments on the Quality of English Language:
English is OK although the manuscript would benefit from an English native speaker read through.
Response: We agree with the comment of the reviewer. Author Services of MDPI will be contacted after completing the revision to enhance the writing quality of the revised manuscript.
Reviewer 2 Report
Comments and Suggestions for Authors
Study with a subject of great interest
Introduction
A distinction of the problem by age, sex, and social stratum is missing.
There is no indication of the specific guidelines that have been taken, fundamentally in exercise, nor the target population to which it is directed; this should be reflected.
The objectives and hypotheses of this study should be clear and explicit.
Methods
It does not reflect how to solve the fact that a self-administered questionnaire may have errors, such as the subject not being able to understand concepts about their health status or the meaning of BMI itself, which may not be accessible to all. This methodology does not mention physical activity.
The target age should also be narrowed to obtain a more realistic view.
Discussion
A sample of 798 people is probably too small to explain the current problems in Saudi Arabia; therefore, its scope should be justified.
Since the numbers are small, it is suggested that the tone of the findings be moderated slightly, as they cannot be generalized.
The limitations of this study include the fact that there were no age or sex classifications used.
Author Response
The author of the manuscript appreciates the important comments of the reviewer. Below are point by point responses to all raised comments. We believe that the applied modifications as per the instructions of the reviewer have enhanced the writing quality of the manuscript.
Comment 1: Study with a subject of great interest
Response: The author of the manuscript appreciates the supportive comment of the reviewer.
Introduction
Comment 2: A distinction of the problem by age, sex, and social stratum is missing.
Response: We agree with the comment of the reviewer concerning the importance of assessing body weight status by age, sex, and social stratum. The current analysis involves bivariate analysis assessing the associations between gender, age, and social stratum with weight regain. Due to the limited sample size of the current investigation, dividing the sample into more strata might lead to loss of power and presence of multiple empty cells (values of zero) within the applied statistical analysis. This is indicated as a weakness in the last paragraph of the discussion section and highlighted with yellow as the following:
‘The current study has a relatively limited sample size, due to the nature of the research question which only targeted those who attempted to lose weight during the last three years leading to exclusion of many participants and subsequently limiting the application of certain statistical tests (such as stratification by age, gender, and social stratum). Therefore, future studies are recommended to include larger sample and to involve other regions in the country.’
Comment 3: There is no indication of the specific guidelines that have been taken, fundamentally in exercise, nor the target population to which it is directed; this should be reflected.
Response: The author of the manuscript appreciates the comment of the reviewer. More details are now added to the data collection tool section of materials and methods to indicate specific guidelines used to develop the data collection tool and similar literature assessing weight loss measures in the targeted population. The following is added to the revised manuscript within the materials and methods section and highlighted with yellow:
‘Content validity was assured through a review of relevant literature concerning weight loss measures in a Saudi Arabian context. The Saudi Guidelines for Prevention and Management of obesity were consulted to identify weight loss measures as per the recommended healthcare practices. Additionally, similar literature assessing weight loss methods in the region of Jazan was also consulted to identify weight loss methods prac-ticed in the same community, which might be considered harmful or less healthier op-tions for weight loss. The compiled list of weight loss measures included exercising, consuming less amount of food, intermittent fasting, selection of low calorie food items, use of artificial sweeteners. consulting a nutritionist or a physician, use of medications associated with weight loss, purging, or bariatric surgery [20, 29].’
Comment 4: The objectives and hypotheses of this study should be clear and explicit.
Response: We appreciate the comment of the reviewer. The objectives and hypotheses are modified as per the request to enhance the writing quality of the manuscript. The following was added to the last paragraph of the introduction and highlighted with yellow:
‘ The rising prevalence of overweight and obesity, along with the corresponding weight loss efforts among adults, underscores the need to evaluate weight loss strategies. Additionally, it is crucial to understand the factors linked to weight management and the prevention of weight regain. Jazan region is one of the regions affected by the rise of excess weight in the country where the prevalence of overweight and obesity reached 47.6% among adults [28]. The increased proportion of individuals with excess weight might motivate some to adopt weight loss measure and might also be at risk of weight regain. This study explores weight loss strategies among adults in Jazan, Saudi Arabia, and assesses the factors contributing to weight regain in this group.’
Methods
Comment 5: It does not reflect how to solve the fact that a self-administered questionnaire may have errors, such as the subject not being able to understand concepts about their health status or the meaning of BMI itself, which may not be accessible to all. This methodology does not mention physical activity.
Response: The author of the manuscript agrees with the comment of the reviewer. Due to the nature of the utilized assessment tool which is based on the recall of the participant, measurement bias induced by this subjective assessment is possible. This was acknowledged as a limitation of the study in the last paragraph of the discussion. The section is highlighted with yellow in the revised manuscript as the following:
‘A significant limitation of the study is its subjective nature, primarily influenced by the participants’ perspectives, rather than associating findings with specific clinical measures. However, it can be argued that this approach is valuable, as only a small proportion of this community typically seeks professional healthcare when trying to lose weight.’
Comment 6: The target age should also be narrowed to obtain a more realistic view.
Response: We appreciate the comment of the reviewer. The current study assessed weight loss attempt and risk of weight regain among adult above the age of 18 without specific age limitation for exclusion criteria. This notion was applied throughout the recruitment process to ensure provision of a sample representative of all age groups. The mean age of the participants was 32 indicating good distribution of younger and older individuals.
Discussion
Comment 7: A sample of 798 people is probably too small to explain the current problems in Saudi Arabia; therefore, its scope should be justified.
Response: We agree with the comment of the reviewer that a larger sample size would be more beneficial for the provision of more generalizable findings. Although the current sample was able to detect multiple statistically significant associations, we also acknowledge the potential impact of the smaller sample size on the study’s findings. The following was added to the last paragraph of the discussion to indicate this as a weakness area and a potential for future research. This section is highlighted with yellow in the revised manuscript as the following:
‘Another limitation is related to the generalizability of the current findings since it only targeted one region in Saudi Arabia. However, it can be postulated that similar cultural, and socioeconomic background in all region of Saudi Arabia is likely to impact lifestyle choices, including weight modification measures. The current study has a relatively limited sample size, due to the nature of the research question which only targeted those who attempted to lose weight during the last three years leading to exclusion of many participants and subsequently limiting the application of certain statistical tests (such as stratification by age, gender, and social stratum). Therefore, future studies are recommended to include larger sample and to involve other regions in the country.’
Comment 8: Since the numbers are small, it is suggested that the tone of the findings be moderated slightly, as they cannot be generalized.
Response: We agree with the comment of the reviewer. The tone is moderated and the impact on the generalizability is now emphasized in the revised version of the manuscript.
Comment 9: The limitations of this study include the fact that there were no age or sex classifications used.
Response: We acknowledge this limitation and is now emphasized and highlighted with yellow in the last paragraph of the discussion within the revised manuscript as the following:
‘The current study has a relatively limited sample size, due to the nature of the research question which only targeted those who attempted to lose weight during the last three years leading to exclusion of many participants and subsequently limiting the application of certain statistical tests (such as stratification by age, gender, and social stratum). Therefore, future studies are recommended to include larger sample and to involve other regions in the country.’
Round 2
Reviewer 1 Report
Comments and Suggestions for Authors
The author has made a reasonable effort in addressing reviewer's comments.